# Structural and biochemical characterisation of the *Providencia stuartii* arginine decarboxylase shows distinct polymerisation and regulation

Matthew Jessop [1,2], Karine Huard[1], Ambroise Desfosses [1], Guillaume Tetreau [1], Diego Carriel[1], Maria Bacia-Verloop[1], Caroline Mas [1], Philippe Mas[1], Angélique Fraudeau[1], Jacques-Philippe Colletier[1] & Irina Gutsche [1✉]

Bacterial homologous lysine and arginine decarboxylases play major roles in the acid stress response, physiology, antibiotic resistance and virulence. The *Escherichia coli* enzymes are considered as their archetypes. Whereas acid stress triggers polymerisation of the *E. coli* lysine decarboxylase LdcI, such behaviour has not been observed for the arginine decarboxylase Adc. Here we show that the Adc from a multidrug-resistant human pathogen *Providencia stuartii* massively polymerises into filaments whose cryo-EM structure reveals pronounced differences between Adc and LdcI assembly mechanisms. While the structural determinants of Adc polymerisation are conserved only in certain *Providencia* and *Burkholderia species*, acid stress-induced polymerisation of LdcI appears general for enterobacteria. Analysis of the expression, activity and oligomerisation of the *P. stuartii* Adc further highlights the distinct properties of this unusual protein and lays a platform for future investigation of the role of supramolecular assembly in the superfamily or arginine and lysine decarboxylases.

[1] Institut de Biologie Structurale, Univ Grenoble Alpes, CEA, CNRS, IBS, 71 Avenue des martyrs, F-38044 Grenoble, France. [2] Present address: Division of Structural Biology, The Institute of Cancer Research (ICR), London, UK. ✉email: irina.gutsche@ibs.fr

nducible AAT-fold basic amino acid decarboxylases (LAOdcs) have been studied since the early 1940s[1,2] because of the link between their enterobacterial pathogenicity in humans and their ability to overcome the aggressive pH environments and oxygen limitation in our gastrointestinal tract. These enzymes decarboxylate either Lysine (LdcI, also called CadA), Arginine (Adc or AdiA) or Ornithine (OdcI or SpeF) into corresponding polyamines while consuming protons and producing $CO_2$, thereby buffering the bacterial cytosol and the extracellular medium to promote bacterial survival under acid stress conditions[3]. In contrast, their acid stress-unrelated counterparts, LdcC, LdcA and OdcC (or SpeC) produce polyamines that regulate bacterial physiology, biofilm formation, virulence and antibiotic resistance[4].

An exhaustive phylogenetic investigation of proteobacterial LAOdcs revealed their separation in two monophyletic groups, OdcIC and LAdc[4]. A further analysis, combining structural and phylogenetic information, partially disclosed relationships among the LAdc subfamilies, with a monophyly of LdcIC and recombination between LdcA and Adc within structural domain boundaries[5]. The genetic environment of *adc* was shown not to be conserved, and the Adc subfamily turned out to be heavily impacted by horizontal transfer and possibly by hidden paralogy, which complicates functional predictions. Although structure-function relationships of Ldcs from many different bacterial species have been largely studied by our group and by others[5–7], the current understanding of Adcs is mainly based on two closely related enterobacterial species, namely *Escherichia coli* and *Salmonella enterica* serovar Typhimurium. In these species, LdcI is induced and active at a broad range of moderately low pHs, whereas Adc offers a robust protection against extreme acid stress down to a pH of approximately 2[3,8]. Excitingly, we have recently observed a patchy distribution of LdcI under the inner membrane of the acid-stressed *E. coli* cell, and showed that in vitro under these conditions LdcI polymerises into filamentous stacks[9]. Indeed, more and more enzymes, particularly in eukaryotes, have been shown to self-assemble into filaments upon environmental stress[10,11]. However, to our knowledge no other LAOdc have been observed to form stacks, and although we and others thoroughly analysed the oligomerisation behaviour of *E. coli* Adc as a function of pH[12,13], no polymerisation could be detected. Since studies of Adcs of more distant enterobacterial species are lacking, we set out to investigate an Adc from a member of the *Proteeae* tribe, *Providencia*, at the periphery of the *Enterobacteriaceae* family composed of over 30 genera.

The reasons for choosing *Providencia* were manifold. First, while *Providencia* are part of normal human gut flora, they are opportunistic multidrug resistant pathogens commonly causing gastrointestinal disturbances in hospitals and nursing homes, and are also isolated from more severe infections such as meningitis[14]. The species of primary significance to human health are *P. stuartii* and to a lesser extent *P. rettgeri*. Second, some clinical isolates of this genus contain urease, an enzyme that breaks urea down into $CO_2$ and ammonia, which can be readily protonated into $NH_4^+$, resulting in pH increase. However, the urease activity was shown to be generally too weak to explain the observed alkalinisation of urine[15,16]. Third, our earlier work showed that the genomes of *Providencia* genus lack *ldcs* but possess an *adc*[4]. Here we solve the cryo-electron microscopy (cryo-EM) structure of the *P. stuartii* Adc decamer to 2.45 Å resolution and demonstrate its high similarity with its *E. coli*[17] and *Salmonella* Typhimurium[18] homologues. The pH-dependency of the arginine decarboxylase activity of these three enzymes is also similar. Remarkably however, in contrast with the *E. coli* and *Salmonella* Typhimurium Adcs, *P. stuartii* Adc self-assembles into stacks of decamers in vitro. We solve the structure of these polymers to a

resolution of 2.15 Å, compare them with the structure of the LdcI stacks[9] and show that the two assembly mechanisms are structurally very different. In addition, the expression pattern of *P. stuartii* Adc is unexpectedly different and dependent on the bacterial social traits. This raises exciting new questions about the roles and evolution of LAOdc polymerisation and opens up novel research perspectives.

## Results

**Structural characterisation of the *P. stuartii* Adc decamer by cryo-EM.** The structure of the *P. stuartii* Adc decamer (~180 Å in diameter) was solved by cryo-EM with an overall resolution of 2.45 Å (Fig. 1a, b, Supplementary Figure 1, Table 1, EMDB: 13261, PDB: 7P9B). The monomer of *P. stuartii* Adc is 758 amino acids long and displays 70.95% and 70.29% sequence identity to the *E. coli* Adc (PDB: 2VYC)[17] and *Salmonella* Typhimurium Adc (PDB: 5XX1)[18] respectively. In the decameric double ring, two monomers form a C2-symmetric dimer (Fig. 1c), which represents the basic unit of the structure and forms a decameric assembly through strong inter-ring contacts between adjacent dimers (Fig. 1a). This type of dimerisation is typical for proteobacterial LAOdcs and for pyridoxal 5′-phosphate (PLP)-dependent enzymes in general, and leads to a completion of the active sites of each monomer buried into a cleft at the dimer interface (Fig. 1c). Like all proteobacterial LAOdcs, the *P. stuartii* Adc monomer is organized in three different structural parts (Fig. 1a). The N-terminal wing domain protrudes into the centre of the double ring and is involved in stabilisation of the decamer assembly though inter-dimer contacts (residues 1-139). Preceded by a short linker region (residues 140-192), the central core part contains the covalently-bound PLP cofactor and is composed of a PLP-binding domain (residues 193-439) and an aspartate aminotransferase-like domain (residues 440-608). The C-terminal domain (residues 609-758), located at the decamer periphery, partially constitutes an entry channel into the active site. The Root-Mean-Square Deviation between *P. stuartii* and *E. coli* Adc monomers is 0.52 Å (over 4424 atoms), which illustrates their strong overall similarity. At the sequence level, the wing domains of the *P. stuartii* and the *E. coli* Adc proteins are the most different (52.14% identity) whereas the PLP-binding domain is the most conserved (77.33% identity). Although the exact residues involved in both dimerisation and decamerisation of the *P. stuartii* and *E. coli* Adcs differ, the buried surface area (BSA) and predicted ΔG upon decamerisation is similar for both proteins (see Supplementary Data 1).

As for the *E. coli* and *Salmonella* Typhimurium Adcs[13,19], *P. stuartii* Adc activity peaks at pH 5 and sharply drops off at pHs above and below (Supplementary Figure 2); the optimal activity of the three enzymes is comparable. The *E. coli* Adc decamer was previously documented to be stable at least between pH 3.5 and pH 5, and to massively dissociate into inactive dimers at pH 6 and above[12,13]. In contrast, our Mass Photometry analysis of *P. stuartii* Adc indicates the presence of the decameric species at pH 5 and 6.5, whereas lower molecular weight species are detected at pH 8 but also unexpectedly at pH 4 (Fig. 1d). Previous analysis of the solvent-accessible charged residues in the *E. coli* Adc crystal structure suggested that low pH promotes association of dimers into decamers through neutralisation of acidic residues in the wing domains[17]. This mechanism implies that the enzyme acts in the acid stress response not only through the decarboxylation reaction that it catalyses but also through the direct binding of protons to its surface under acidic conditions. Interestingly, both the total number of charged residues and proportion of charged residues at the surface of the *P. stuartii* decamer are higher than for *E. coli* Adc which may have implications for a pH-dependent

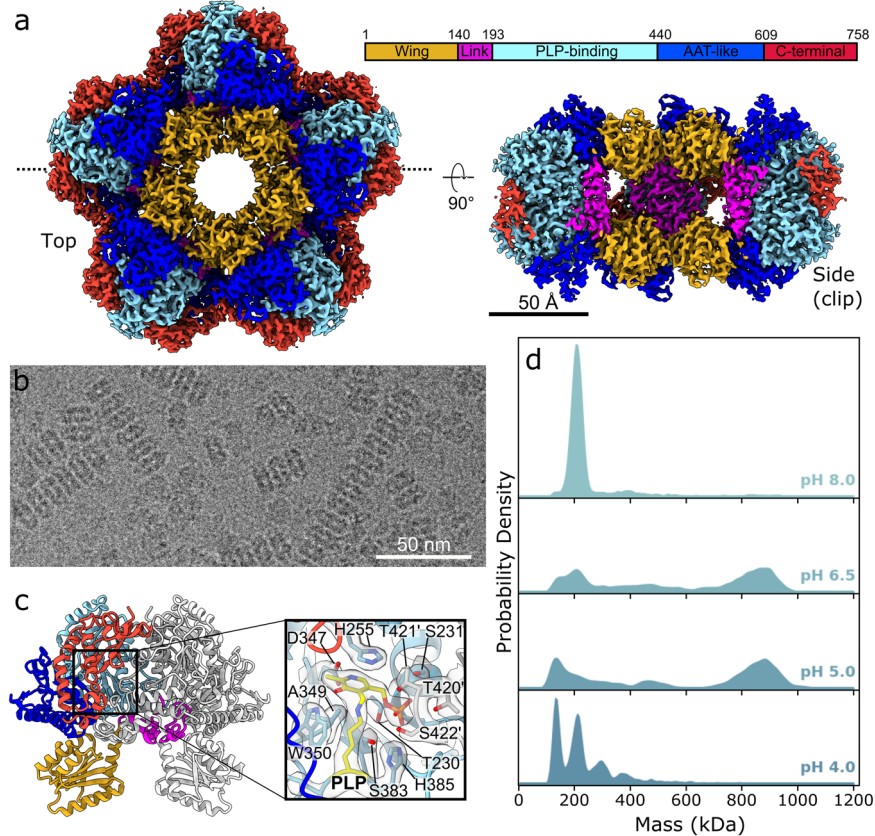

**Fig. 1 Characterisation of the P. stuartii Adc decamer. a** Cryo-EM map of the *P. stuartii* Adc decamer. Top and clipped side views of the 2.4 Å resolution cryo-EM map are shown, with the clipping plane indicated by dotted lines on the top view. Domains are coloured by domain as shown in the schematic, with amino acid numbers at domain boundaries corresponding to the first residue of each domain. **b** Crop of a representative micrograph of *Providencia stuartii* Adc, scale bar = 50 nm. **c** Cartoon representation of a *P. stuartii* Adc dimer extracted from the decamer structure. One monomer in the dimer is coloured by domain as per **a**, the other is coloured light grey. The inset shows a close up of the active site, with active site residues shown as sticks and labelled. The PLP cofactor covalently bound to K386 is shown in yellow. The cryo-EM density for active site residues is overlaid. **d** Mass photometry characterisation of *P. stuartii* Adc oligomerisation showing the distribution of species of different molecular weights at pH 4.0, 5.0, 6.5 and 8.0.

regulation of this enzyme. For a reader interested in more extensive exploration of the distribution of charged residues and prediction of the pH-induced changes in this distribution, see Supplementary Data 1 and Supplementary Figure 3 for a detailed analysis that highlights the exciting complexity of these purportedly well-understood assemblies.

**Structural determinants of the P. stuartii Adc polymerisation into linear stacks revealed by cryo-EM.** In particular starting from pH 6.5 and below, our mass photometry movies indicated the presence of an increasing number of large species outside the usable mass range (see Methods). Moreover, as we observed *P. stuartii* Adc by cryo-EM with the goal to solve the structure of the decamer presented above, we discovered that most of the Adc double rings were linearly stacked to create double-rings and longer filaments (Figs. 1b, 2). Consequently, we solved the structure of the *P. stuartii* Adc stacks at pH 6.5 that was used for protein purification to a resolution of 2.15 Å (Fig. 2a, Supplementary Figure 4, Table 1). The atomic model of the decamer stack reveals that upon polymerisation the conformation of the Adc decamer remains extremely similar, with minor changes localised to interfaces between monomers and dimers, and at the interface between decamers (Supplementary Data 1). Thus, under these conditions, the decamer structure is primed for filament formation. The inter-decamer interface is mainly stabilised by a salt bridge between D472 and R492, both located in the AAT domain and respectively situated in a loop following helix α18

and in a β-hairpin. Interestingly, these residues are highly conserved in *P. stuartii* and *P. rettgeri* (Supplementary Data 2), suggesting a strong evolutionary pressure towards Adc polymerisation in these species. Remarkably, D472 and R492 are replaced by G472 and T492 respectively in the solved structures of *E. coli* and *Salmonella* Typhimurium Adcs, providing an explanation for the inability of these enzymes to polymerise (Fig. 2). We note that the isoelectric point of the decamer shifts to 6 upon stacking, in line with the burial of acidic residues at the stack interface (Supplementary Data 1). Calculation of the effect of pH on the per-monomer free energy of folding suggests that, at pH higher than 4, stack formation is only slightly more favoured than that of isolated decamers (Supplementary Data 1 and Supplementary Figure 7). Thus, it is not surprising that these two forms coexist at pH 6.5 used for cryo-EM imaging. These observations, combined with the presence of stacks at all, but in particular at acidic, pHs, the high structural similarity between the free and the stacked decamers at pH 6.5, the very similar mass photometry profiles at pH 6.5 and pH 5, and the activity peak at pH 5, suggest that the decamers and the stacks represent the active form of the *P. stuartii* Adc enzyme.

**Evolutionary conservation of the structural determinants of Adc polymerisation.** Our previous survey of AAT-fold LAOdcs led to their clear separation into three clusters[4]: (i) Cluster I with wing-less LAOdcs mainly found in Firmicutes, Cyanobacteria and Actinobacteria, (ii) Cluster II containing a few Firmicute

**Table 1 Cryo-EM data collection, refinement and validation statistics.**

| | Adc stack EMDB 13466 PDB 7PK6 | Adc decamer EMDB 13261 PDB 7P9B |
|---|---|---|
| Data collection and processing | | |
| Magnification | 215,000 | 215,000 |
| Voltage (kV) | 300 | 300 |
| Electron exposure (e−/Å²) | 40.0 | 40.0 |
| Defocus range (μm) | −0.5-−2.5 | −0.5-−2.5 |
| Pixel size (Å) | 0.65 | 0.65 |
| Symmetry imposed | D5 | D5 |
| Initial particle images (no.) | 438,429 | 303,475 |
| Final particle images (no.) | 268,579 | 46,854 |
| Map resolution (Å) | 2.15 | 2.44 |
|  FSC threshold | 0.143 | 0.143 |
| Map resolution range (Å) | 2.07–3.56 | 2.42–3.29 |
| Refinement | | |
| Initial model used (PDB code) | 2VYC | PDB 7PK6 |
| Model resolution (Å) (masked) | 2.42 | 2.56 |
|  FSC threshold | 0.5 | 0.5 |
| Map sharpening B factor (Å²) | −74 Å⁻² | −75 Å⁻² |
| Model composition | | |
|  Non-hydrogen atoms | 120,580 | 60,290 |
|  Protein residues | 15,100 | 7550 |
|  Ligands | 0 | 0 |
| B factors (Å⁻²) | | |
|  Protein | 23.00 | 47.08 |
| R.m.s. deviations | | |
|  Bond lengths (Å) | 0.006 | 0.009 |
|  Bond angles (°) | 0.649 | 1.127 |
| Validation | | |
|  MolProbity score | 1.47 | 1.51 |
|  Clashscore | 4.28 | 7.23 |
|  Poor rotamers (%) | 2.22 | 0.93 |
| Ramachandran plot | | |
|  Favoured (%) | 98.00 | 97.47 |
|  Allowed (%) | 2.00 | 2.53 |
|  Outliers (%) | 0.00 | 0.00 |

sequences and nearly all proteobacterial sequences of winged LAOdcs, and (iii) a small Cluster III with a mix of winged and wing-less sequences from unrelated taxa. For phylogenetic analyses of Clusters I and II, one species per strain was then selected randomly to limit taxonomic redundancy. This curated Cluster II contained in particular 69 sequences of proteobacterial Adcs and 56 sequences of LdcIs[4]. Thus, now that we saw that D472 and R492 are involved in *P. stuartii* Adc polymerisation, the availability of these thoroughly aligned sequences enabled us to straightforwardly evaluate their conservation in proteobacterial Adcs. In addition to *P. stuartii*, this analysis revealed conservation of both D472 and R492 in the Adcs of three betaproteobacterial species containing LdcA rather than LdcI – *Burkholderia cepacia*, *B. multivorans* and *B. vietnamiensis* - with which it shares around 49% sequence identity (Supplementary Figure 5). Noteworthy, these species are opportunistic pathogens causing life-threatening infections in patients with cystic fibrosis or other immunocompromising diseases. This unexpected conservation suggests that Adcs of these species may also be capable of stack assembly.

**Comparison between structural determinants of *P. stuartii* Adc and *E. coli* LdcI stack formation reveals very different assembly mechanisms**. The differences between decamer structures of LdcI and Adc are translated into the structures of the polymers formed by stacking of double-ring building blocks (Fig. 3a, b). In fact, a direct visual comparison of cryo-EM 2D class averages of *P. stuartii* Adc and *E. coli* LdcI stacks already reveals striking differences between the two polymers (Supplementary Figure 6a), the foremost distinguishing feature being the compactness of the inter-decamer packing. Whereas the LdcI decamers are tightly nested into inter-dimer grooves of neighbouring decamers (PDB: 6YN6), the Adc decamers appear to be less tightly packed, which is reflected by the BSA upon decamer stacking being $6108 \pm 7.3$ Å² for LdcI but only $1609 \pm 2.1$ Å² for *P. stuartii* Adc (Supplementary Data 1). Specifically, although the diameters of the LdcI and Adc decamers are very similar, the height of the Adc decameric ring is 105 Å compared to the 90 Å height of the LdcI decamer (Supplementary Figure 6b). This height difference, resulting in a 95 Å repeat distance between successive *P. stuartii* Adc stacked decamers compared to 76 Å in the LdcI stacks (Fig. 3a, b), is caused by an inserted β-hairpin in Adc (Fig. 3), which contains the stack-forming residue R492, as well as by an approximately 10° rotation in the wing domain (Supplementary Figure 6b, c).

Closer analysis of the inter-decamer interfaces in the two stacks reveals the differences in the mechanism of stack formation. For consistency with our previous manuscript[9], the inter-decamer interfaces contributing to the *E. coli* LdcI stack formation are annotated as interface 1 and interface 2 (Fig. 3b). Remarkably, despite the clear differences in the arrangement of decamers in Adc and LdcI stacks, the stack interface in *P. stuartii* Adc is situated at a similar location to interface 2 in LdcI (Supplementary Figure 6d). Whereas interface 2 in LdcI is formed by reciprocal end-on interactions at the end of helix α16, the three-amino acid extension of α18 in Adc (equivalent in position to α16 in LdcI), coupled with the inserted β-hairpin, causes a rotation between decamers of 20°. This contrasts to the LdcI stack structure, where there is negligible rotation between neighbouring decamers in the stack (Supplementary Figure 6a). Thus, instead of a reciprocal end-on interaction between opposing α18 helices, D472 at the end of α18 interacts with R492 from the inserted β-hairpin (Fig. 3b, Supplementary Figure 6d). Each interface between two dimers across the decamer interface is made of two such interactions due to the dihedral symmetry of the decamer.

Notably, whilst only two residues are involved in Adc stacking (with a possible contribution of a S64-S64 interaction), interface 1 between *E. coli* LdcI decamers in the polymer is formed by eight resides (N314, D316, E344, G352, R468, D470, E482, Y485) and interface 2 by six (N94, R97, T444, E445, S446, D447). Excitingly, in 22 out of the total of 56 curated Cluster II LdcIs, at least 10 out of these 14 residues are conserved, and all these species belong to the *Enterobacteriaceae* family (Supplementary Table 1, Fig. 3e). In addition, the *E. coli* LdcI H694 that we previously identified as being involved in the pH-dependent switch that induces polymerisation of *E. coli* LdcI upon acid stress appears to be very conserved even beyond *Enterobacteriaceae*. Thus, polymerisation of LdcI during the Enterobacterial stress response is likely to be a general phenomenon, whereas polymerisation of Adcs may be restricted to several *Providencia* and *Burkholderia* species.

Finally, it should be noted that the optimal expression and activity of the *E. coli* LdcI is achieved at around pH 5.7, and that while the stacking propensity of LdcI at pH below 6 is very high, no stack formation was so far detected above this pH[9,13,20]. This agrees with our prediction of the charge distribution and free energy of folding as a function of pH (Supplementary Data 1). Superposition of the dimer structures extracted from the isolated LdcI decamer at pH 7.0 and from the stack at pH 5.7 revealed an inter-monomer rotation resulting in a narrowing of the central

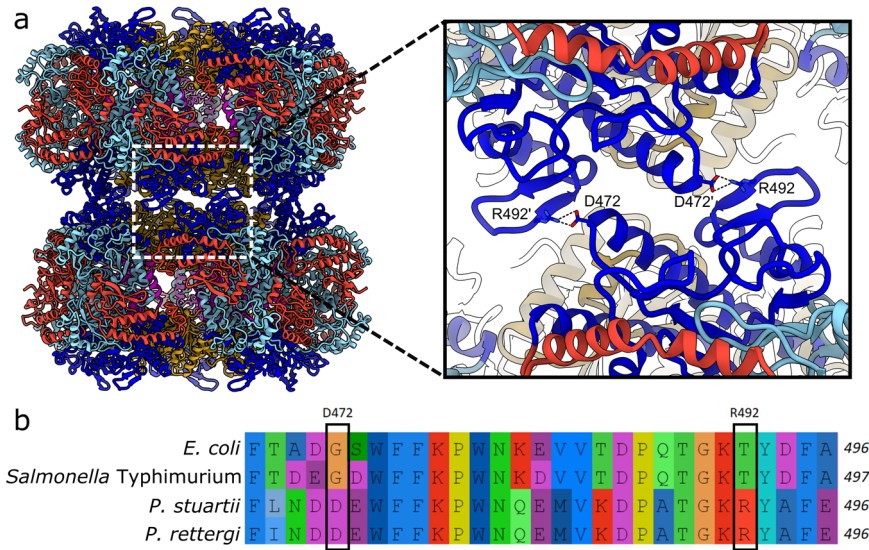

**Fig. 2 Molecular determinants of Adc stack formation. a** Atomic model of a 2-decamer stack shown as cartoons, with domains coloured as in Fig. 1a. The inset shows a closeup of the decamer:decamer interface in the stack, with the key stack-forming residues D472 and R492 shown as sticks. **b** Multiple sequence alignment of Adc sequences from *E. coli* (WP_001381593.1), *Salmonella* Typhimurium (WP_000978690.1), *P. stuartii* (WP_004923888.1) and *P. rettgeri* (WP_164564359.1) focused on region around key stack-forming residues D472 and R492 in the *P. stuartii* Adc (indicated by boxes), coloured according to the AliView colour scheme.

cavity and positioning of interface residues in the decamer to trigger polymerisation[9]. In the present work however, both the Adc decamer and Adc stack structures were solved at pH 6.5, and, as specified above, the decamers polymerised as essentially rigid entities, notwithstanding small changes at the dimerisation and decamerisation interfaces (Supplementary Data 1). Comparison with the *E. coli* and the *Salmonella* Typhimurium Adcs, which were crystallised at pH 6.5 and pH 8.0 respectively, does not reveal compelling pH-dependent rearrangements. In addition, in the case of the *P. stuartii* Adc we did not observe any clear cut-off pH of the stack formation, but noticed increased precipitation of the protein at lower pHs, which may tentatively be explained by massive stack assembly. This hypothesis is supported by our calculations of the free energy of folding, which suggests a maximum gain in free energy upon stack formation at pH 4 (Supplementary Data 1, Supplementary Figure 7). In line with the tighter packing, larger BSA, more elaborate interaction network, and generally much longer polymers of the *E. coli* LdcI as compared to the *P. stuartii* Adc (see Fig. 1b of the present manuscript and Fig. 4b in ref. [9] for a visual comparison), Supplementary Figure 7 and Supplementary Data 1 highlight yet another interesting aspect of the mechanistic differences in the polymerisation of the two enzymes: in particular at low pHs, the change in the free energy of folding upon LdcI decamer stacking is around two times higher than upon Adc stacking. We previously proposed that protonation state of H694 buried at the dimer interface may influence the *E. coli* LdcI polymerisation propensity at different pHs. However, we could not identify any obvious candidate for a pH-dependent switch in the Adc stack structure. The equivalent residue in the *P. stuartii* Adc is E739, which is involved in an intra-chain salt bridge with K256, suggesting a distinct mechanism of the *P. stuartii* Adc regulation in comparison with both *E. coli* Adc and LdcI.

**Adc expression is affected by pH and ammonia but not by urea and is dependent on *P. stuartii* socialisation status.** In *E. coli* and *Salmonella* Typhimurium, the Adc system expression is maximal at pH lower than 5, under anaerobic conditions and in the presence of its substrate arginine. *P. stuartii* however was shown to be viable only between pH 5 and pH 9[16]. This corresponds to the acidity in its pathophysiological habitat, the human urinary tract, a challenging environment enriched in urea (up to 170 mM) and its metabolite ammonia (up to 25 mM)[21]. Noteworthy, these substances are both products of arginine catabolism. The higher the pH, the stronger the tendency of *P. stuartii* to attach to surfaces, for example to medical devices such as catheters in a hospital setting, thereby forming resilient biofilms. These are able to cope with a wide variety of environmental stresses, including antibiotic treatment, causing severe infections[22]. Furthermore, even in its planktonic form, *P. stuartii* is already highly social and forms floating communities of cells (FCC) with increased resistance that precede and/or coexist with the canonical surface-attached biofilms (SAB), probably facilitating the host colonisation process through dispersion[16]. Since transition from a planktonic to biofilm lifestyle is generally known to modulate bacterial genome expression, we addressed the expression of the *P. stuartii adc* gene by quantitative reverse transcription quantitative PCR (RT-qPCR) in both lifestyles over a pH range of pH 5 to pH 9 and exposed to different concentrations of urea and ammonia (Fig. 4).

In contrast to *E. coli* and *Salmonella* Typhimurium, *adc* expression was detected at all pHs (Fig. 4a). A statistically significantly higher *adc* expression level was observed in SAB compared to FCC (Supplementary Table 2). In SABs, expression gradually increases with the pH, with a 2.6-fold higher expression at pH 9 than at pH 5. This unexpected pH-dependence is not observed in FCC, in which the expression is significantly lower at neutral than at more extreme pHs. The presence of urea at concentrations up to 1 M does not significantly affect *adc* expression in both lifestyles at pH 7, and *adc* is consistently 4.7 to 12.9-fold more expressed in SAB compared to FCC when exposed to urea. In contrast, increasing ammonia concentrations boost the *adc* gene expression both in the FCC (up to 11.2-fold) and in the SAB (up to 3.2-fold). In summary, in addition to the structural and biochemical particularities of the *P. stuartii* Adc, this investigation of the *adc* expression pattern reveals further differences in the *adc* gene regulation pattern compared to the *E. coli* and *Salmonella* Typhimurium Adcs.

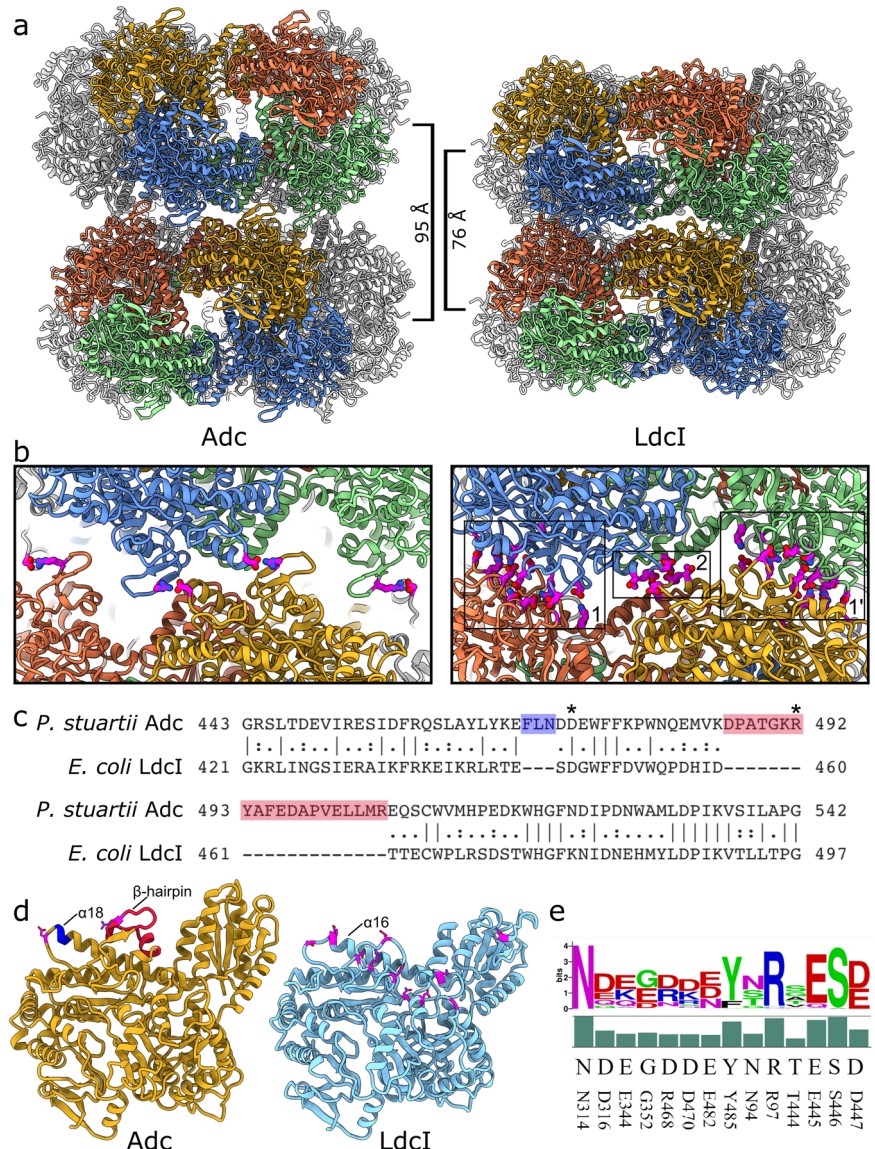

**Fig. 3 Comparison between Adc and LdcI stack formation. a** Atomic model of two-decamer Adc and LdcI stacks. The closer packing of decamers in the LdcI stack reduces the repeat distance between decamers in the stack from 95 Å to 76 Å. **b** Close-up view of the decamer interfaces in the Adc stack and LdcI stack. Stack-forming residues are shown in magenta as sticks. The decamer:decamer interface in the Adc stack is equivalent to interface annotated '2' in the LdcI stack, with an offset due to the inserted β-hairpin in Adc. This insertion, along with a 3-amino acid insertion at the end of α-helix α18, has the effect of pushing the decamers further apart in Adc with the consequence that there is no interface in Adc equivalent to interface '1' in LdcI, reducing the total buried surface area between two decamers from 6108 Å$^2$ to 1609 Å$^2$. **c** Pairwise sequence alignment of LdcI and Adc in the beginning of the AAT-like domain, with insertions in Adc highlighted in green and red. The key stack-forming residues in Adc D472 and R492 are marked with an asterisk. **d** Monomeric structures of LdcI and Adc, with stack-forming residues shown in magenta as sticks. The Adc insertions highlighted in **c** are shown on the Adc structure in the same colours. **e** Conservation of the 14 residues involved in interfaces 1 and 2 of the LdcI stack formation across the Cluster II LdcIs represented as a sequence logo. The bar chart indicates the relative frequency of the consensus sequence residues. Residue numbers are from the *E. coli* LdcI sequence.

## Discussion

Altogether, our data highlight a number of surprising differences between the *P. stuartii* Adc and its well-characterised *E. coli* and *Salmonella* Typhimurium homologues. The expression profile of the *P. stuartii* *adc* exemplifies the uniqueness of the gene regulation in this bacterium, and further illustrates the dependence of its stress responses on the bacterial social traits. The higher *adc* expression in the SAB than in the FCCs seems counter-intuitive considering that the free-swimming planktonic colonies are supposedly more exposed to environmental stresses than the sessile biofilms. Interestingly, a similar trend has been previously observed for expressions of genes coding for *P. stuartii* porins[16].

The physiological role of the *P. stuartii* Adc is also enigmatic, because although its optimal enzymatic activity sharply peaks at pH 5 as for the *E. coli* and *Salmonella* Typhimurium Adcs, the *P. stuartii* bacterium is not viable at pHs below 5, and the pH inside the bacterial cytoplasm is never expected to reach this value. Akin to this, the enhanced *adc* expression in the SAB upon increasing pH is surprising, firstly, because the *E. coli* and *Salmonella* Typhimurium Adcs are extreme acid stress response enzymes[3], and secondly, because even though the pH inside the biofilm is less basic than in the surrounding medium, at the optimal expression conditions the enzyme is supposed to be inactive and partly dissociated into lower oligomeric species. In this respect,

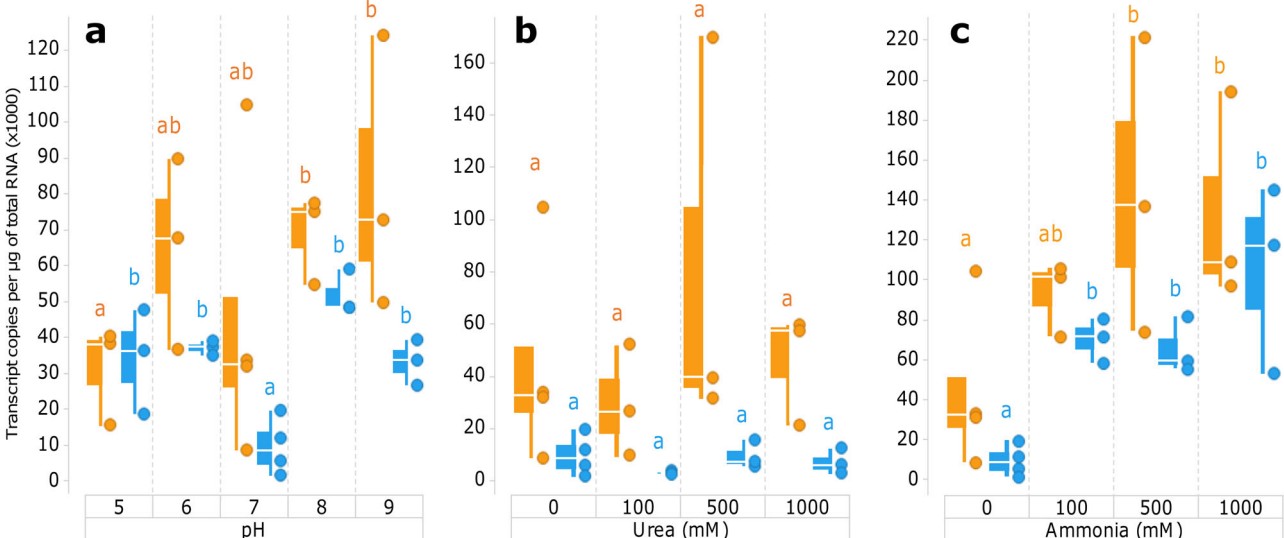

**Fig. 4 Expression of the adc gene quantified by RT-qPCR in floating colonies and biofilms as a function of pH, urea and ammonia.** For each bacterial lifestyle, conditions within each environmental insult were statistically compared pairwise as illustrated by letters above the corresponding distributions (blue for FFC and orange for SAB). Only letters of the same colours (referring to the same phenotype) can be compared. For each colour, when all letters are different between two distributions, then the expression levels in the two corresponding conditions are considered significantly different (Post-Hoc Tukey HSD; $p < 0.05$). If two distributions share at least one letter, then the expression levels in these conditions are not significantly different with a $p \geq 0.05$. **a** Exposure of *P. stuartii* to different pHs. **b** Exposure to increasing concentrations of urea. **c** Exposure to ammonia.

similarly to the *E. coli* LdcI[9], the ability of the *P. stuartii* Adc to polymerise into stacks may provide an elegant means to locally increase its concentration and maintain its activity. The *P. stuartii* Adc may also act as a buffer via neutralisation of its highly negatively charged surfaces and function as a biosynthetic rather than an acid stress response enzyme. Indeed, the Adc reaction product agmatine is further catabolized by agmantine ureohydrolase into the most common bacterial diamine, putrescine, and urea, which is then hydrolysed by urease releasing two ammonium molecules and $CO_2$. In parallel, enterobacterial putrescine is directly synthesised by Odc via ornithine decarboxylation, and is a source for synthesis of other polyamines, such as spermidine. Both putrescine and spermidine are well-known players in bacterial biofilm formation and numerous other essential processes[23,24]. Specifically, *Providencia* putrescine has been directly implicated in the bacterial swarming motility[25] which is of great importance to the colonisation process[26]. Phenotypical analysis of chromosomal mutants of *P. stuartii* with impaired Adc expression, enzymatic activity or polymerisation propensity should shed light on the mechanism of action of this protein. Our current work provides a solid basis for investigations of the role of the *P. stuartii* Adc in general and the stack forming residues in particular. As a first step, to validate the molecular determinants of the *P. stuartii* polymerisation in vitro, beyond the high quality of the cryo-EM map and the resulting atomic model, D472 and/or R492 could now be mutated into alanine to abolish the decamer stacking, and the D472-R492 salt bridge could be swapped into R472-D492.

The presence of straight and ordered *P. stuartii* Adc filaments in the cryo-EM images is remarkable in itself because whereas LdcI polymerisation was observed in negative stain EM as early as 1974[27], such behaviour has not yet been described for an Adc in spite of intense investigation. Indeed, our sequence alignments show that while in certain *Providencia* and *Burkholderia* species D472 and R492 are conserved, it is not the case for most of the other proteobacterial Adcs. Thus, we suppose that Adc polymerisation confers specific advantages to these bacteria under certain conditions and might therefore represent a gain of

function. For example, Adc assembly into stacks may potentially contribute to a spatial organisation of polyamine biosynthesis in these species or be used to inform the cell of the level of arginine or other environmental stimuli. Considering the difference in the structures of LdcI and Adc decamers and in the geometry of their stacking, polymerisation mechanisms of these enzymes seem to have evolved independently. In this regard, an extensive analysis of point mutations triggering protein polymerisation demonstrated that amplification of protein interactions by dihedral symmetry enhances the polymerisation probability, and highlighted the universal prevalence of hydrophobic residues at the assembly interfaces[28]. Specifically, this study evaluated the frequency of amino acid presence at the polymer interfaces versus protein surfaces, and defined a "stickiness" scale, with hydrophobic residues being the most "sticky" and charged residues such as lysine, glutamate and aspartate the least "sticky" residues, with a protective effect against protein self-association. Interestingly however, although the predicted polymerisation-inducing mutations D460L in *E. coli* LdcI (PDB: 3N75) and K491Y in *E. coli* Adc (PDB: 2VYC) resulted in an in vivo fibre formation upon expression in yeast and the D460L LdcI mutant was shown to form filaments in vitro, no self-assembly of the purified K491Y Adc mutant could be observed[28]. Furthermore, our cryo-EM structures of the native *E. coli* LdcI and *P. stuartii* Adc stacks show that the interdecamer interfaces are formed nearly exclusively by charged and polar residues (Supplementary Data 1, Supplementary Table 1), and that these mutations are not directly located in the identified interfaces. Thus, the assumption that evolutionary sampling of protein polymerisation occurs primarily via hydrophobicity-increasing mutations may not be universal. For example, prevalence of charged and polar interactions may facilitate the regulation of filament formation in a pH or ionic strength-dependent manner, as proposed for instance for the yeast enzymes glutamine synthetase[29] and CTP synthase[30]. Regardless of the exact molecular mechanism, the number of enzymes discovered to polymerise into filaments with purposes as diverse as enzymatic activity regulation, scaffolding, controlling of cell shape and signalling, is constantly growing[11]. Remarkably,

although more than two dozen linear metabolic enzyme polymers have already been characterized by electron microscopy or X-ray crystallography, most of them form helices and only five, including *E. coli* LdcI and now *P. stuartii* Adc, are dihedrally-symmetric rings that polymerise by ring stacking[11]. Understanding the advantages of enzyme assembly into filaments for their function and/or regulation both in vitro and in vivo is an exciting topic for future studies. Our present results on *E. coli* LdcI and *P. stuartii* Adc should undoubtedly stimulate further investigation and influence the way one analyses the LAOdc superfamily that, despite being studied for 80 years, appears to still hold many mysteries.

## Methods

**Cloning, expression and purification.** The *adc* gene (Gene accession from genome of *P. stuartii* strain ATCC 33672: NZ_CP008920.1:1566120-1568396) was amplified by PCR from the genomic DNA of the *Providencia stuartii* ATCC 29914 strain (Pasteur Institute, Paris, France) and inserted into the pET22b(+) vector instead of the *E. coli ldcI* gene[6] using the Gibson cloning strategy. The primers ADCPst_F1g (5′-AGATATACATATGAGGGCACTAATTGTTTATACCGAGC-3′) and ADC Pst_R1g (5′-CCGAATTGATTTCGTCACATGTTTTCACACACATCACA-3′) were used for the *adc* gene amplification and the primers pET22b_F1g (5′-TGT GACGAAATCAATTCGGATCCCCATGGC-3′) and pET22b_R1g (5′-AGTGCC CTCATATGTATATCTCCTTCTTAAAGTTAAACAAAATTATTTCTAGAGG GG-3′) for the plasmid backbone amplification.

The Adc protein containing a TEV cleavable 6X His-tag sequence was overproduced in BL21 DE3 cells (Novagen) in LB medium supplemented with ampicillin at 37 °C, upon overnight induction with 0.5 mM IPTG at 18 °C. Cell pellets were resuspended in a 25 mM Tris-HCl, 300 mM NaCl, 0.1 mM PLP, 10% glycerol, pH 7.5 buffer supplemented with Complete EDTA free (Roche), and subsequently disrupted by microfluidizer at 4 °C. Clarified supernatant was loaded on a Ni-NTA column and eluted with 500 mM imidazole. After extensive dialysis to remove the imidazole, the protein was further concentrated and purified by size-exclusion chromatography on a Superose 6 column in a 25 mM MES, 300 mM NaCl, 0.1 mM PLP, 1 mM DTT, 10% glycerol, pH 6.5 buffer.

**Cryo-EM data collection and 3D reconstruction.** Purified Adc was diluted to a final concentration of approximately 0.25 mg/mL in a buffer containing 25 mM MES, 150 mM NaCl, 1 mM DTT and 0.1 mM PLP, pH 6.5. 3 µL of the sample was applied to a glow-discharged R2/1 300 mesh holey carbon copper grid (Quantifoil Micro Tools GmbH) and plunge-frozen in liquid ethane using a Vitrobot Mark IV (FEI) operated at 100% humidity at room temperature. Datasets were recorded at the European Synchrotron Radiation Facility (ESRF) in Grenoble, France[31], on a Titan Krios microscope (Thermo Scientific) equipped with a BioQuantum LS/967 energy filter (Gatan) and a K2 summit direct electron detector (Gatan) operated in counting mode. A total of 10,023 movies of 40 frames were collected with a total exposure time of 3 s, total dose of 40 e−/Å2 and a slit width of 20 eV for the energy filter. All movies were collected at a nominal magnification of 215,000 x, corresponding to a pixel size of 0.65 Å/pixel at the specimen level. A summary of cryo-EM data collection parameters can be found in Table 1. Micrographs were manually screened based on the presence of particles, amount of contamination and apparent beam-induced movement, resulting in a total of 7037 selected micrographs. Image processing was performed in cryoSPARC[32] as follows.

For the filament structure, an initial reference was obtained by manually picking a small subset of filaments, followed by ab initio reconstruction and refinement in cryoSPARC using D5 symmetry. This reference was used to generate reference projections for template picking, limiting out-of-plane tilt to 20 degrees, and resolution of the templates to 20 Å. The minimum distance between picks was set to 60 Å, so that one particle per stack in the filaments was picked. The resulting 438,429 particles were extracted in 576 pixels boxes and iteratively classified in 2D in order to keep only series of three decamers, paying attention to avoid single or double decamers. This resulted in 268,579 particles which were refined against the above-mentioned initial model filtered at 20 Å, using D5 symmetry, per-particle defocus refinement and default cryoSPARC refinement options, leading to a 2.3 Å resolution map. Given the relative long-range disorder of the filaments, an additional round of 3D refinement was performed with a tighter mask that enclosed only the central decamer and the inter-decamer interfaces. This resulted in a final map at an estimated resolution of 2.15 Å (FSC 0.143), which was sharpened with a B-factor of −74 Å2 for visualization and model building.

For the single decamer structure, we masked out the central decamer from the filament structure in order to generate reference projections for template picking, filtered at 15 Å. The minimum distance between picks was set to 220 Å, therefore limiting the number of picked decamers within filaments. Following extraction in 512 pixels boxes, the resulting 303,475 particles were extensively 2D-classified in order to keep only single, isolated decamers. Due to the inherent imprecision of 2D classification, at each selection round, the non-selected particles were re-classified to retrieve a maximum number of particles. The resulting set of 46,854 particles

was refined using D5 symmetry, per-particle defocus refinement and default cryoSPARC refinement options, giving a final map at an estimated resolution of 2.4 Å (FSC 0.143), which was sharpened with a B-factor of −75 Å2 for visualization and model building.

Local resolution and FSC curves are shown in Supplementary Figure 1 and Supplementary Figure 4 for the decamer and the stack maps respectively.

**Fitting of structures and refinement.** A homology model of the *P. stuartii* Adc monomer was first generated using the Phyre2 server[33], using the X-ray crystallography structure of *E. coli* Adc (PDB 2VYC) as a reference. The homology model was then rigid-body fitted in Chimera[34] into the cryo-EM map of the *P. stuartii* three-decamer stack, and then D5-symmetrised to form a decameric ring. The decamer was duplicated and then rigid-body fitted to form a model of a two-decamer stack; only two decamers were required to characterise the decamer-decamer interface. After a round of manual corrections in Coot[35], a first round of real-space refinement was carried out in PHENIX 1.13-2998[36] with rigid-body, global minimization, local grid search and ADP refinement parameters enabled, and imposing rotamer, Ramachandran and NCS restraints (chain A as the reference for all other chains). After a second round of manual correction in Coot, two further rounds of refinement were carried out in Phenix, with the 'nonbonded_-weight' parameter first set to 500 and subsequently to 100.

For the single decamer model, one decamer from the two-decamer stack model was rigid-body fitted into the map in Chimera. A single round of real-space refinement in Phenix was carried out with rigid-body, global minimization, local grid search and ADP parameters enabled, with the 'nonbonded_weight' parameter set to 100. The resulting model was then manually inspected and corrected in Coot, with very few adjustments needed. Validation for both the stack and decamer models was then carried out in Phenix 1.71.1 using the comprehensive validation (cryo-EM) job. A summary of refinement and model validation statistics can be found in Table 1.

**Structure analysis.** Buried surface area and interactions at the dimer, decamer and stack interfaces were computed and identified using PISA[37]. For the prediction of surface charge distributions as a function of pH, we used PROPKA[38] to titrate charged residues, protonated the structures using PDB2PQR[39] with the amber force field[40]. APBS[41] was then used to perform Poisson-Boltzmann electrostatics calculations and generate the electrostatic surfaces. Free energies of folding of the dimers, decamers and stacks were computed by PROPKA and normalized for comparison by subtraction of the highest energy value (corresponding to the unfolded chains at pH 14), thereby setting the maximum energy to zero, and division by the number of dimers in the considered oligomers (i.e., 1, 5 or 10 for the isolated dimers, isolated decamers and stacked decamers, respectively). The underlying assumption is that regardless of the starting state, all protein chains are unfolded at pH 14 (set as zero kcal/mol), meaning that the free energy is then proportional to the number of chains and can be compared. Hence we subtracted, for each protein in each state, the free energy calculated at pH14 from those calculated at the other pH, and then divided the resulting value by the number of chains in the oligomers. This treatment enables eliminating the 'noise' contribution to calculated free energies arising from non-perfect agreement between the experimental structures geometries and the force field (Amber), which scales up proportionally with the number of chains in the oligomers. Figures were prepared with Pymol[42].

**Mass Photometry data acquisition and analysis.** Mass Photometry experiments were carried out using a Refeyn OneMP (Refeyn Ltd., Oxford, UK) MP system. AcquireMP and DiscoverMP software packages were used to record movies and analyse data respectively using standard settings. Microscope coverslips (high precision glass coverslips, Marienfeld) were cleaned following the Refeyn Ltd Individual rinsing procedure. Reusable self-adhesive silicone culture wells (Grace Bio-Labs reusable CultureWell™ gaskets) were used to keep the sample droplet shape. Contrast-to-mass calibration was carried out using a mixture of proteins with molecular weights of 66, 146, 480 and 1048 kDa. Immediately prior to the measurements, protein stocks were diluted directly in stock buffers at pH 4.0, pH 5.0, pH 6.5 and pH 8.0 to reach concentrations of 10 to 40 nM. To this end, 1 to 2 µL of protein solution was added into 18 to 19 µL of analysis buffer, to reach a final drop volume of 20 µL. The final Adc concentrations used in the experiments presented in Fig. 1d were 10 nM at pH 8.0, 20 nM at pH 6.5, 30 nM at pH 5.0 and 40 nM at pH 4.0. The reason for the usage of increasing protein concentrations with decreasing pH was that the lower the pH, the more frequently the mass photometry movies contained very large entities, presumably corresponding to Adc stacks. Thus, the total sample concentration was increased in an attempt to partly compensate for the presence of these entities, beyond the 3 to 5 MDa upper limit of reliable mass photometry measurements according to the manufacturer's specifications, and only the lower molecular weight species (i.e., the decamer and below) were further analysed with DiscoverMP.

**Isothermal titration calorimetry.** Isothermal titration calorimetry was used to assess the activity of the enzyme at different pHs as described for Ldcs[5,6]. All experiments were carried out using a MicroCal iTC200 instrument (Malvern) at

20 °C. To obtain the reaction enthalpy of the conversion of L-arginine to agmatine by Adc, single injection mode experiments were conducted using one injection of 40 μl 8 mM L-arginine over a period of 400 sec into 25 nM Adc, followed by 1500 s spacing. The single injection mode experiments were performed at pH 4, pH 5, pH 6.5 and pH 8 in reaction buffers containing 100 mM NaCl, 0.1 mM PLP, 10% glycerol and either 100 mM acetic acid-sodium acetate (for the pH 4 buffer and pH 5 buffers), 100 mM MES (for the pH 6.5 buffer), or 100 mM Tris-HCl (for the pH 8 buffer). Considering that a measurable Adc activity was detected at pH 5 only, the determination of the enzymatic parameters was conducted in the pH 5 buffer. 16 injections of 2.5 μL 8 mM L-arginine were applied 100 s apart, into 10 nM Adc. The experiment was performed in triplicate, and data was analysed using the Michaelis-Menten equation and a nonlinear regression model within the R software[43] using the nlstools package[44]. Normality of the variable distribution was confirmed with the Shapiro-Wilk's test ($p$-value = 0.3456). The enzymatic parameters within the confidence interval between 2.5% and 97.5% were estimated as follows: $K_M =$ 0.331 mM (0.28, 0.38) and $k_{cat} = 102$ s$^{-1}$ (97, 108).

While in particular plant Adcs decarboxylate both arginine and ornithine, bacterial bifunctional wing domain-containing AT-fold Adcs have to our knowledge not been described[45]. Thus, a potential L-ornithine decarboxylation activity of the *P. stuartii* Adc was not assessed in this work.

**Analysis of Adc and LdcI sequence alignments**. Adc and LdcI sequences were extracted from the curated aligned LAOdc Cluster II sequences[4] and conservation of the stack-forming residues evaluated. Residues corresponding to the *E. coli* LdcI and *P. stuartii* Adc stack forming interfaces were used for building sequence Logo[46] and consensus sequence with WebLogo Software (https://weblogo.berkeley.edu/). Jupyter Notebooks and Python Libraries (Pandas, Numpy & Matplotlib) were used for sequence analysis and data visualisation.

Adc sequences from *P. stuartii* and *P. rettgeri* were obtained by using BLAST algorithm[47] with default parameters (Expect Threshold: 0,05, Word size: 6, Matrix: BLOSUM62, Gap Costs: Existence: 11, Extension: 1). Alignments were examined with AliView[48].

**Adc expression in *P. stuartii***. The *adc* expression in conditions mimicking the environmental insults experienced by *P. stuartii* SAB and FCC was quantified by RT-qPCR. The *P. stuartii* ATCC 29914 strain was cultured by adding 1 colony in 30 mL of LB medium and incubating 1 h at 37 °C and 150 rpm agitation. Three biological replicates were performed for each condition, consisting in three wells of a 6-well plate (CytoOne) filled with 2 mL of the culture and incubated at 37 °C under 60 rpm agitation. FCC in suspension were collected and briefly centrifuged to collect the bacteria without damaging them. FCC were challenged by resuspending in either 2 mL of fresh LB containing urea or NH₄Cl (ranging from 100 to 1000 mM for both), or in 2 mL of LB medium with pH adjusted between 5 and 9, and incubated at 37 °C for 30 min under 60 rpm agitation. In parallel, SAB remaining attached to the plate well bottom was washed with LB before adding fresh LB modified as described above. The plate was incubated at 37 °C for 30 min under 60 rpm agitation.

RNA was extracted from each condition by using the RNeasy Mini Kit (Qiagen) with some adjustments for the two different phenotypes in the primary steps. For FCC, 1 mL of the 2-mL culture was removed and replaced by 1 mL of RNAprotect Cell Reagent (Qiagen). After 5 min of incubation, the mixture was centrifuged at 3,200 g for 10 min and the pellet was resuspended in 110 μL of TE buffer (Tris-HCl 30 mM, EDTA 1 mM, pH 8.0) containing 15 mg/mL of lysozyme and 2 mg/mL of proteinase K (Qiagen), and incubated at RT for 10 min. For SAB, the 2 mL of LB medium were removed from well and 1 mL of RNAprotect Cell Reagent: LB (1:1) was added. After 5 min incubation, the RNAprotect was removed and 410 μL of TE buffer containing 15 mg/mL of lysozyme and 0.5 mg/mL of proteinase K were added. To recover the maximum of SAB, the adherent biofilm was mechanically disrupted by scratching the well bottom. Once resuspended, the SAB was incubated at RT for 10 min. All remaining steps of the extraction procedure were performed following manufacturer's instructions. Total RNA was quantified for each condition and each phenotype Nanodrop2000 (Thermo Fisher Scientific). RNA was immediately used for reverse transcription. 1 μg of RNA was reverse transcribed into complementary DNA (cDNA) by using the QuantiTect Reverse Transcription Kit (Qiagen) following manufacturer's instructions. cDNA was stored at −20 °C until use and the remaining RNA were stored at −80 °C.

Absolute quantification by RT-qPCR was used to precisely quantify *adc* expression over the set of conditions tests. A 95 bp fragment of *adc* was amplified using the primers ADCPst1_F1q (5′-TTAATCAAGCCTATCTAATGC-3′) and ADCPst1_R1q (5′-ACTATTGCCATCCATCAT-3′) and subsequently used as a standard using eight concentrations from $5.0 \times 10^1$ to $5.0 \times 10^8$ gene copies. Each reaction was constituted of 4 μL of ten-fold diluted samples (or of the standards), 0.6 μL of each primer (final concentration of 10 μM each), 7 μL of SsoAdvanced Universal SYBR Green Supermix (Bio-Rad) and RNase-free water for a final reaction volume of 15 μL in each well. For each of the three biological replicates of each condition, two technical replicates were done. The qPCR was run on a CFX Connect Real-Time PCR Detection System (Bio-Rad) using the following programme: an initial step at 95 °C for 3 min, then 39 cycles of 95 °C for 10 sec (denaturation) and 60 °C for 45 s (primers annealing & elongation). The specificity of the amplification was assessed by performing a melt curve analysis. The number of transcript copies of *adc* was calculated using the standard curve by using Bio-

Rad CFX Manager v.3.1 (Bio-Rad). Gene expression data was normalized using the total RNA content for each condition, as it is the preferable normalization method when no housekeeping gene has been validated in the tested species, which is the case in *P. stuartii*[49].

Statistical analyses were conducted to investigate the effect of different parameters on *adc* gene expression using the R software. First, a generalized linear model (GLM) analysis was performed to assess, in each environmental insult (exposure to different pH, ammonia or urea), the effect of the parameters 'condition' (*i.e.*, FCC or SAB) and 'dose' (*i.e.*, the pH value or the concentration of ammonia or urea). Considering that an effect of each or both of these parameters was observed, an ANOVA followed by a post-hoc test (Tukey's HSD) was further conducted to perform pairwise comparisons of *adc* expression levels between conditions. Bartlett test and Shapiro-Wilk tests confirmed the homoscedasticity and normality of variance distribution, respectively, supporting the use of an ANOVA for such comparisons.

**Statistics and reproducibility**. Data was collected on independent experiments. At least three independent experiments were run for each condition. All statistical analyses were performed using the R software, version 3.3.2. Normal distribution was tested using the Shapiro–Wilk normality test.

**Reporting summary**. Further information on research design is available in the Nature Research Reporting Summary linked to this article.

## Data availability

The datasets generated during the current study are available from the corresponding author on reasonable request. Cryo-EM maps, along with the corresponding fitted atomic structures, have been submitted to the EMDB and PDB with accession codes EMD: 13261 and PDB: 7P9B for *P. stuartii* Adc decamers and EMD: 13466 and PDB: 7PK6 for Adc stacks. Source data for Figs. 1d and 4 can be found in Supplementary Data 3 and Supplementary Data 4 respectively.

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

## Acknowledgements

We thank Guy Schoehn for establishing and managing the cryo-electron microscopy platform and for providing training and support, Daphna Fenel and Emmanuelle Neumann for assistance at the EM platform, Alister Burt for help with the Mass Photometry figure and Jan Felix for help with the model refinement and stimulating discussions. We acknowledge the European Synchrotron Radiation Facility for provision of beam time on CM01, and are grateful to Eaazhisai Kandiah for the final cryo-EM data acquisition. This work was funded by the European Union's Horizon 2020 research and innovation programme under grant agreement No 647784 to IG. The research was also supported by the Agence Nationale de la Recherche (grant ANR-2018-CE11-0005-02 to JPC). We used the platforms of the Grenoble Instruct-ERIC centre (ISBG; UMS 3518 CNRS-CEA-UGA-EMBL) within the Grenoble Partnership for Structural Biology (PSB), supported by FRISBI (ANR-10-INBS-05-02), and GRAL, financed within the University Grenoble Alpes graduate school (Ecoles Universitaires de Recherche) CBH-EUR-GS (ANR-17-EURE-0003). The electron microscope facility is supported by the Auvergne-Rhône-Alpes Region, the Fondation Recherche Médicale (FRM), the fonds FEDER, and the GIS-Infrastructures en Biologie Santé et Agronomie (IBISA).

## Author contributions

M.J., K.H., A.D., G.T., D.C., M.B.V., C.M., P.M., A.D., J.P.C. and I.G. performed experiments and analysed data. I.G. designed the overall study, supervised the project and wrote the paper with significant input of M.J. and contributions from all authors.

## Competing interests

The authors declare no competing interests.
