## [Peer Review File · Communications Biology]

Reviewers' comments:

Reviewer #1

The manuscript "Structural and biochemical characterization of the *Providencia stuartii* arginine decarboxylase – a unique enterobacterial decarboxylase with distinct polymerization and regulation" by Jessop et al reports special properties of arginine decarboxylase (ADC) that are not found with other ADCs. Many bacteria can survive at very acidic conditions because of the presence of lysine decarboxylase (LDC) and ADC. ADCs are large proteins that form 522 symmetric decamers at low pH. LDCs also form decamers. However, in contrast to ADCs, the decamers of LDCs polymerize to form long clusters. ADC decamers do not usually polymerize to larger molecular complexes. The authors demonstrate that *Providencia stuartii* ADC, unlike ADCs from other bacterial sources, polymerize into large molecular complexes by the stacking of decamers as in LDCs. However, the stacks of ADC decamers are distinct from stacks of LDC decamers as the separation of decamers in ADCs is significantly larger than that of LDCs. The authors provide molecular reasons for decamer formation as well as stacking of decamers. The authors follow standard experimental protocols for electron microscopic reconstruction as well as biochemical and biophysical studies. *Providencia stuartii* ADC forms decameric structures at pH 5.0 and pH 6.5 at which the decamer is catalytically active. At pH 4.0 and pH 8.0, the molecule disassembles to lower molecular weight aggregates which are not enzymatically active. Interestingly at pH 4.0, four different molecular weight species are observed. Also, at pH 5.0 and pH 6.5, apart from the decameric species, lower molecular weight species comparable to the species at pH 8.0 are observed. The species at pH 8.0 seems to be homogenous. The explanation given for these observations is not entirely satisfactory. Is it possible to get better insights? The manuscript is well written. The authors use 'cryo-EM map' for describing the structure derived from cryo-EM map. I am not sure if this is appropriate. The inset in Figure 1C is supposed to show cryo-EM density for a bound PLP. It is not possible to visualize the density. The figure clearly needs to be improved.

In summary, the manuscript is a valuable contribution to the literature on the structural characteristics of *Providencia stuartii* ADC and could be accepted for publications after a minor revision.

Reviewer #2 (Remarks to the Author):

This is a truly high impact paper, addressing fundamental, frontline issues in the biological sciences, namely the nature of protein-protein recognition, ultimately leading to 'supramolecular assembly' (as very nicely highlighted at a high level in the discussion in reference 26 – the 2017 Nature report by Levy and co-workers). The structures reported here are quite novel and they raise very interesting questions about function, particularly given the 'inverse' pH dependence seen on the expression of the title enzyme (more below).

This study by Gutsche and co-workers describes the cryo-EM structure of L-arginine decarboxylase (ADC) from *P. stuartii*, solved to quite high resolution, 2.15 Å, particularly considering the nature of the quaternary structure (decameric-stacked pentagonal structure). The work builds on recently solved structures of bacterial L-lysine decarboxylase (LDC) and other bacterial ADCs from *E. coli* and *S. typhimurium*.

And while the pH dependency of ADC activity for these three ADCs is similar, the authors report here one striking difference among these enzymes. Namely, similar to the very exciting work reported by the Gutsche team earlier this year (PNAS, 2021, ref. 9) in the case of the inducible LDC, the Grenoble team reports here that the *P. stuartii* ADC oligomerizes in a stack-like manner (Figure 3 provides a very nice comparison). The repeat distance in the stack is 93 Å for ADC vs. a more compact 76 Å for the inducible LDC.

There will be great interest in the details of the supra-structures determined here and in the mechanism by which they self-assemble and in their biological role. While these questions cannot be definitively answered herein, the information gleaned from such studies helps us actually to more precisely formulate such questions, and lays a beautiful groundwork for their future investigation.

Questions for the Authors:

(1) What is the distribution of stack lengths (oligomeric size distribution) seen here with the *P. stuartii* ADC and how does this compare with the LDC-inducible stacks seen earlier by the Gutsche team?

(2) The authors observe a fascinating R492-D-472' D472-R492' electrostatic-pairing interactions seen in *P. stuartii* and *P. rettgeri*--but not in *E. coli* or *S. typhimurium* ADC—are critical for the donut stacking interactions seen here. Please expand upon this. What are the likely energetics of these interactions and are they consistent with the expected affinity of one decamer to another (ideally one would want to be able to determine or estimate a decamer-decamer dissociation constant here)? This is also very different from what is typically seen in protein-centric supramolecular assembly that is usually triggered by increasing surface hydrophobicity as is discussed in reference 26 – please place the observations here in the context of what is known about protein-self assembly in this regard.

(3) The authors note that whereas the bacterial inducible LDC is (over)expressed in response to acid stress, and the filament-like structures that are seen perhaps serve as a mechanism to deploy, localize and concentrate the acid-neutralizing capability of bacterial LDC, this is not what is seen with the *P. stuartii* ADC, the topic of this manuscript. Namely, *P. stuartii* ADC expression is elevated at higher pH! The authors suggest that this enzyme may serve more of a signaling role, controlling flux into the polyamine pathway (agmatine to putrescine, then on to spermidine and spermine?). Please discuss the role of polyamines in bacterial function. Would a filamentous structure or a fibrous ADC confer an advantage for localizing diamine or polyamine biosynthesis?

(4) Related to question (3), many so-called ADC enzymes actually decarboxylate both arginine and ornithine, giving either agmatine – that can be hydrolyzed to putrescine- or putrescine directly! What are the properties of the *P. stuartii* ADC, in this regard? Does it give putrescine directly by decarboxylation of ornithine?

Typos:

- First sentence: Change 'homogous' to 'homogeneous'
- Reference 8 – typo on name of second author
- Change 'S. Typhimurium' to 'S. typhimurium' throughout
- Figure 3: There is a discrepancy between the stacking distance for *P. Stuartii* ADC in the figure itself (95 Å) and in the legend (93 Å) – please fix so that this is inconsistent. Perhaps, adding experimental uncertainty will do this – should this be 93 ± 2 Å?

Reviewer #3 (Remarks to the Author):

In this manuscript Jessop and colleagues report the cryoEM structure of the arginine decarboxylase (Adc) from the human pathogen *Providencia stuartii* (*P. stuartii*) at around 2.4 Å resolution as decamer and around 2.1 Å resolution for a higher-order polymer. The authors highlight unique features of this enzyme by comparing Adc to other bacterial decarboxylases. The most pronounced differences between the *Escherichia coli* (*E. coli*) lysine decarboxylase LdcI and *P. stuartii* Adc can be found in the polymer stacking mechanisms of decamers when forming higher-order structures. While LdcI (as described previously) forms compact polymers with a

relatively short decamer repeat distance, packing of Adc into polymers is more loosely organised with a relatively small inter-decamer interface. This small inter-decamer interface mainly consists of multiple specific salt bridges formed by an aspartate residue D472 of one monomer and an arginine residue R492 of another monomer. Consequently, this interface-forming salt bridge is used to predict filament formation in other species. The authors conclude their study with comparing gene expression patterns of Adc in different bacterial systems.

This study beautifully describes the differences in polymer assembly and uses multiple sequence alignments to draw conclusions about the general appearance of different polymer assemblies in bacterial species, including filament formation of Adc. The paper is written in an excellent style and the figures are almost completely self-explanatory and well designed. The study is bolstered by careful bioinformatician analysis and overall well designed, which is why I can support publication of this work in Communications Biology.

However, when reading this paper two obvious main questions seems not to be addressed yet, which I believe would significantly strengthen the authors conclusions drawn in this manuscript and therefore should be addressed (if feasible). Even if not all the questions can be biochemically addressed, an explanatory comment in the manuscript would be useful for the reader.

Major:

- The authors describe in detail the relatively small interface formed by pairs of residues in the inter-decamer interface and make predictions which species have the capability to form filamentous structures.

Several biochemical experiments could be envisioned to further confirm these findings.

(1) Perhaps most important would be to mutate these residues and perform negative stain and photometry experiments to confirm that when mutating those residues, polymer structures formation is indeed abolished.

Possible mutant could include a double mutation of D472 and R492 to alanine or a swap in charge by mutating only R492 to glutamate.

The importance of this salt bridge could be confirmed by swapping the charge of both residues, namely D472R and R492D.

Lastly, proteins that are predicted to not form polymers could be transformed into filament-forming polymers by introducing this specific interface. However, such gain-of-function experiments might indeed not be feasible and are not required.

- My second main question links structure and function of the protein. *P. stuartii* Adc self-assembles into stacks of decamers in vitro under certain pH conditions, however from this study it does not seem clear to me if these decamers will indeed be functional or not?

TOROID structures (cited by the authors) have been shown to functional inactive the TORC1 activity and might therefore execute a storage function. Is that comparable to such filaments? Can the authors separate filaments from lower-molecular weight structures and compare the activity of each species?

Minor:

1. The authors use mass photometry in Fig. 1d to show the distribution of high molecular weight assemblies at different pH conditions. This could be correlated with negative stain microscopy to validate the photometry results.

2. In Fig. 1c, residues in the active site and the PLP cofactor should be labelled to guide the reader.

3. In Fig. 4, the meaning of "a" and "b" above the error bar is unclear and should be explained in the figure legend.

4. "Bacteria homologous lysine and arginine decarboxylases...". I believe the authors meant homologous in the abstract.

5. Reference 8 contains two question marks in the author name.

6. In the figure legend of Fig. 1, 1e is mentioned that probably existed in earlier versions of the manuscript but no longer exist.

7. Fig. 3c may benefit from a real close-up view to compare side chain-side chain interactions in the different assembly modes.

Reviewer #1

The manuscript “Structural and biochemical characterization of the *Providencia stuartii* arginine decarboxylase – a unique enterobacterial decarboxylase with distinct polymerization and regulation” by Jessop et al reports special properties of arginine decarboxylase (ADC) that are not found with other ADCs. Many bacteria can survive at very acidic conditions because of the presence of lysine decarboxylase (LDC) and ADC. ADCs are large proteins that form 522 symmetric decamers at low pH. LDCs also form decamers. However, in contrast to ADCs, the decamers of LDCs polymerize to form long clusters. ADC decamers do not usually polymerize to larger molecular complexes. The authors demonstrate that *Providencia stuartii* ADC, unlike ADCs from other bacterial sources, polymerize into large molecular complexes by the stacking of decamers as in LDCs. However, the stacks of ADC decamers are distinct from stacks of LDC decamers as the separation of decamers in ADCs is significantly larger than that of LDCs. The authors provide molecular reasons for decamer formation as well as stacking of decamers.

The authors follow standard experimental protocols for electron microscopic reconstruction as well as biochemical and biophysical studies. *Providencia stuartii* ADC forms decameric structures at pH 5.0 and pH 6.5 at which the decamer is catalytically active. At pH 4.0 and pH 8.0, the molecule disassembles to lower molecular weight aggregates which are not enzymatically active. Interestingly at pH 4.0, four different molecular weight species are observed. Also, at pH 5.0 and pH 6.5, apart from the decameric species, lower molecular weight species comparable to the species at pH 8.0 are observed. The species at pH 8.0 seems to be homogenous. The explanation given for these observations is not entirely satisfactory. Is it possible to get better insights?

This Reviewer’s comment made us realise that the Mass Photometry Figure 1D may convey a wrong impression of the absence of stacks, although they are actually always present, and their relative amount increases with the decreasing pH. Indeed, because the masses of the filaments are beyond the 3 to 5 MDa upper limit of reliable mass photometry measurements according to the manufacturer’s specifications, this particular experiment was only focused on the lower molecular weight species (i.e. the decamer and below), whereas the stacks were investigated by cryo-EM. We now specify in the main text and in the Mass Photometry methods section that very large entities, presumably corresponding to Adc stacks, were frequently observed in the mass photometry movies, but not analysed by the DiscoverMP software because their mass was outside the reliable measurements range.

In addition, we provide numerous tentative explanations for the Adc and LdcI oligomerisation profiles in Supplementary Data 1, Supplementary Figure 3 and Supplementary Figure 7, and all sections throughout the text where we refer to these Supplementary Data and Figures.

The manuscript is well written. The authors use ‘cryo-EM map’ for describing the structure derived from cryo-EM map. I am not sure if this is appropriate.

We have carefully assessed our usage of the term ‘cryo-EM map’ and replaced it by ‘atomic model’ where appropriate.

The inset in Figure 1C is supposed to show cryo-EM density for a bound PLP. It is not possible to visualize the density. The figure clearly needs to be improved.

We have modified the inset to make it clearer, changed the colouring of the PLP moiety and labelled the active site residues, as also suggested by Reviewer 3. The legend to the Figure 1C reflects these modifications.

In summary, the manuscript is a valuable contribution to the literature on the structural

characteristics of *Providencia stuartii* ADC and could be accepted for publications after a minor revision.

Reviewer #2 (Remarks to the Author):

This is a truly high impact paper, addressing fundamental, frontline issues in the biological sciences, namely the nature of protein-protein recognition, ultimately leading to 'supramolecular assembly' (as very nicely highlighted at a high level in the discussion in reference 26 – the 2017 Nature report by Levy and co-workers). The structures reported here are quite novel and they raise very interesting questions about function, particularly given the 'inverse' pH dependence seen on the expression of the title enzyme (more below).

This study by Gutsche and co-workers describes the cryo-EM structure of L-arginine decarboxylase (ADC) from *P. stuartii*, solved to quite high resolution, 2.15 Å, particularly considering the nature of the quaternary structure (decameric–stacked pentagonal structure). The work builds on recently solved structures of bacterial L-lysine decarboxylase (LDC) and other bacterial ADCs from *E. coli* and *S. typhimurium*.

And while the pH dependency of ADC activity for these three ADCs is similar, the authors report here one striking difference among these enzymes. Namely, similar to the very exciting work reported by the Gutsche team earlier this year (PNAS, 2021, ref. 9) in the case of the inducible LDC, the Grenoble team reports here that the *P. stuartii*, ADC oligomerizes in a stack-like manner (Figure 3 provides a very nice comparison). The repeat distance in the stack is 93 Å for ADC vs. a more compact 76 Å for the inducible LDC.

There will be great interest in the details of the supra-structures determined here and in the mechanism by which they self-assemble and in their biological role. While these questions cannot be definitively answered herein, the information gleaned from such studies helps us actually to more precisely formulate such questions, and lays a beautiful groundwork for their future investigation.

Questions for the Authors:

(1) What is the distribution of stack lengths (oligomeric size distribution) seen here with the *P. stuartii* ADC and how does this compare with the LDC-inducible stacks seen earlier by the Gutsche team?

While we did not quantify the distribution of stack lengths of the *P. stuartii* Adc as compared to the *E. coli* LdcI, a visual comparison of the cryo-EM micrographs used for their respective 3D reconstructions shows that the LdcI stacks are much longer and more rigid than the Adc stacks (which agrees with their tighter packing, larger BSA, more elaborate interaction network and the twice as large change in the free energy of folding upon decamer stacking at lower pHs). We now include this in the result section comparing the two types of stack assemblies.

A thorough average length comparison may not bring significant insights in this case because the two enzymes have a different optimal pH for stack formation and the data was collected at different pHs, not corresponding to these respective optima. Moreover, in our experience the length of these filaments in cryo-EM depends to a great extent on the care of their handling during cryo-EM grid preparation (for instance, cutting or not the pipette tip and pipetting from the top or the bottom of the eppendorf tube). In addition, in particular LdcI filaments tend to form also larger bundles, often resulting in too thick ice and unusable for cryo-EM analysis.

(2) The authors observe a fascinating R492-D-472' D472-R492' electrostatic-pairing interactions seen in *P. stuartii* and *P. retergeri*--but not in *E. coli* or *S. typhimurium* ADC—are critical for the

donut stacking interactions seen here. Please expand upon this. What are the likely energetics of these interactions and are they consistent with the expected affinity of one decamer to another (ideally one would want to be able to determine or estimate a decamer-decamer dissociation constant here)?

We did not attempt to experimentally measure the affinity of one decamer to the other in particular because we always observe a mixture of different species in solution (except at pH 8 where the protein is mostly dissociated into dimers, similarly to the *E. coli* Adc and LdcI). The coexistence of different species can be tentatively explained by our calculation of the effect of pH on the per-monomer free energy of folding which shows that, at pH higher than 4, stack formation is only slightly more favoured than isolated decamers (Supplementary Data 1 and Supplementary Figure 7). We expand on the comparative energetics of the *P. stuartii* Adc, *E. coli* Adc and *E. coli* LdcI in Supplementary Data 1.

This is also very different from what is typically seen in protein-centric supramolecular assembly that is usually triggered by increasing surface hydrophobicity as is discussed in reference 26 – please place the observations here in the context of what is known about protein-self assembly in this regard.

We feel, as pointed out by the Reviewer in the first comment, that we already extensively discuss our findings in the context of the reference 26 and the usual prevalence of hydrophobic interactions in the self-assembly interfaces. Since our cryo-EM structures of the *E. coli* LdcI and *P. stuartii* Adc stacks show that the interdecamer interfaces are formed nearly exclusively by charged and polar residues, in the discussion section we now also suggest that the presence of charged and polar interactions in the interfaces may facilitate the regulation of filament formation in a pH or ionic strength-dependent manner, as proposed for instance for the yeast glutamine synthetase and CTP synthase. We provide two additional references to illustrate this proposal.

(3) The authors note that whereas the bacterial inducible LDC is (over)expressed in response to acid stress, and the filament-like structures that are seen perhaps serve as a mechanism to deploy, localize and concentrate the acid-neutralizing capability of bacterial LDC, this is not what is seen with the *P. stuartii* ADC, the topic of this manuscript. Namely, *P. stuartii* ADC expression is elevated at higher pH! The authors suggest that this enzyme may serve more of a signaling role, controlling flux into the polyamine pathway (agmatine to putrescine, then on to spermidine and spermine?). Please discuss the role of polyamines in bacterial function. Would a filamentous structure or a fibrous ADC confer an advantage for localizing diamine or polyamine biosynthesis?

Since we already discussed the roles of bacterial polyamines in our recent publication (Carriel et al., *Genome Biol Evol* 2018), and considering that these roles are extremely diverse and that the present work does not really address the roles of agmatine, putrescine and spermidine in *P. stuartii*, in this revised version we choose to restrict ourselves to providing additional references to excellent recent reviews on this subject in the discussion section.

To our knowledge spatial organisation of polyamine biosynthesis in bacteria have not yet been described. However, this hypothesis seems appealing and would deserve further investigation. Therefore, in the discussion section where we elaborate on a possible gain of function conferred to certain *Providencia* and *Burkholderia* species by the presence of the conserved D472-R492 interaction between Adc decamers in a filament, we now suggest that Adc polymerisation may potentially conduce to a spatial organisation of polyamine biosynthesis in these species or be used to inform the cell of the level of arginine or other environmental stimuli.

(4) Related to question (3), many so-called ADC enzymes actually decarboxylate both arginine and ornithine, giving either agmatine – that can be hydrolyzed to putrescine- or putrescine directly!

What are the properties of the *P. stuartii* ADC, in this regard? Does it give putrescine directly by decarboxylation of ornithine?

While in particular plant Adcs decarboxylate both arginine and ornithine, bacterial bifunctional long AAT-fold Adcs have to our knowledge not been described. The latest publication dedicated to this subject is the very recent work of Li et al., *J Biol Chem* 2021 Oct;297(4):101219. doi: 10.1016/j.jbc.2021.101219. Thus, we assume that the *P. stuartii* Adc can only decarboxylase arginine to agmatine.

Typos:

- First sentence: Change ‘homogous’ to ‘homogeneous’

This typo has been corrected and the length of the abstract reduced from 200 to 147 words to comply with the formatting guide for final submissions.

- Reference 8 – typo on name of second author

This typo has been corrected.

- Change ‘*S. Typhimurium*’ to ‘*S. typhimurium*’ throughout

We are not experts in nomenclature and have therefore referred to the literature to clarify this point. The most up to date reference we could find on *Salmonella* nomenclature/capitalisation is the following <https://link.springer.com/article/10.1007%2Fs12223-011-0075-4>. According to this paper, “serovar names are not italicized but written in Roman type and starts with a capital letter”. In addition, the authors suggest “To simplify the long nature of nomenclature afterwards, the name may be shortened with the genus name followed directly by the serovar such that *S. enterica* subsp. *enterica* serovar Panama becomes *Salmonella* Panama... a serovar name should not follow an abbreviated genus name, e.g. *S. Dublin*”. Therefore, we think that it would be most appropriate to use ‘*Salmonella Typhimurium*’ instead of ‘*S. Typhimurium*’ or ‘*S. typhimurium*’. We modified the text accordingly. We leave the final decision to the Editorial Office.

- Figure 3: There is a discrepancy between the stacking distance for *P. Stuartii* ADC in the figure itself (95 Å) and in the legend (93 Å) – please fix so that this is inconsistent. Perhaps, adding experimental uncertainty will do this – should this be 93 ± 2 Å?

We have corrected the error in the legend for Figure 3A in the reported distance between Adc decamers in the stack – it should be 95 Å as per the Figure. The previous measurement of 93 Å came from an earlier version of the atomic model, 95 Å is the correct distance.

Reviewer #3 (Remarks to the Author):

In this manuscript Jessop and colleagues report the cryoEM structure of the arginine decarboxylase (Adc) from the human pathogen *Providencia stuartii* (*P. stuartii*) at around 2.4 Å resolution as decamer and around 2.1 Å resolution for a higher-order polymer. The authors highlight unique features of this enzyme by comparing Adc to other bacterial decarboxylases. The most pronounced differences between the *Escherichia coli* (*E. coli*) lysine decarboxylase LdcI and *P. stuartii* Adc can be found in the polymer stacking mechanisms of decamers when forming higher-order structures. While LdcI (as described previously) forms compact polymers with a relatively short decamer repeat distance, packing of Adc into polymers is more loosely organised with a relatively small inter-decamer interface. This small inter-decamer interface mainly consists

of multiple specific salt bridges formed by an aspartate residue D472 of one monomer and an arginine residue R492 of another monomer. Consequently, this interface-forming salt bridge is used to predict filament formation in other species. The authors conclude their study with comparing gene expression patterns of Adc in different bacterial systems.

This study beautifully describes the differences in polymer assembly and uses multiple sequence alignments to draw conclusions about the general appearance of different polymer assemblies in bacterial species, including filament formation of Adc. The paper is written in an excellent style and the figures are almost completely self-explanatory and well designed. The study is bolstered by careful bioinformatician analysis and overall well designed, which is why I can support publication of this work in Communications Biology.

However, when reading this paper two obvious main questions seems not to be addressed yet, which I believe would significantly strengthen the authors conclusions drawn in this manuscript and therefore should be addressed (if feasible). Even if not all the questions can be biochemically addressed, an explanatory comment in the manuscript would be useful for the reader.

Major:

- The authors describe in detail the relatively small interface formed by pairs of residues in the inter-decamer interface and make predictions which species have the capability to form filamentous structures.

Several biochemical experiments could be envisioned to further confirm these findings.

(1) Perhaps most important would be to mutate these residues and perform negative stain and photometry experiments to confirm that when mutating those residues, polymer structures formation is indeed abolished. Possible mutant could include a double mutation of D472 and R492 to alanine or a swap in charge by mutating only R492 to glutamate. The importance of this salt bridge could be confirmed by swapping the charge of both residues, namely D472R and R492D.

Lastly, proteins that are predicted to not form polymers could be transformed into filament-forming polymers by introducing this specific interface. However, such gain-of-function experiments might indeed not be feasible and are not required.

The 2.15 A resolution reconstruction of the *P. stuartii* Adc polymer provides confidence in the relevance of the D472-R492 interaction. However, we concur with this Reviewers' comment and agree that would be logical to mutate the residues that we identified as important for the stack formation and assess the effects of the mutation on the protein oligomerisation and activity. We had indeed performed analogous experiments for the *E. coli* LdcI as presented in our PNAS 2021 manuscript. While we are currently unable to perform this mutational analysis on the *P. stuartii* Adc, we now explicitly include a discussion of this point in the discussion section of the manuscript.

- My second main question links structure and function of the protein. *P. stuaritii* Adc self-assembles into stacks of decamers in vitro under certain pH conditions, however from this study it does not seem clear to me if these decamers will indeed be functional or not? TOROID structures (cited by the authors) have been shown to functional inactive the TORC1 activity and might therefore execute a storage function. Is that comparable to such filaments? Can the authors separate filaments from lower-molecular weight structures and compare the activity of each species?

As answered to a comment of Reviewer 2, we observe that the different oligomeric species of *P. stuartii* Adc co-exist in a pH-dependent equilibrium, and cannot be really separated. We tentatively

explain this observation by providing and commenting on our extensive calculations of the distribution of charged residues at the surfaces of each oligomeric species, exploring the predicted pH-induced changes in this distribution, and analysing the effect of pH on the per-monomer free energy of folding of each species (Supplementary Data 1 and Supplementary Figures 3 and 7). We admit that even though this thorough analysis does not offer a complete explanation, this in our view nicely illustrates how complex these types of assemblies are.

As far as the activity of the different species is concerned, because we show that (i) in particular decamers and stacks co-exist at acidic pHs, (ii) the structures of the free and the stacked decamers at pH 6.5 are nearly identical, (iii) the mass photometry profiles at pH 6.5 and pH 5 are also very similar, and (iv) the activity peaks at pH 5, we propose that the decamers and the stacks represent the active form of the *P. stuartii* Adc enzyme. We now include this reasoning in the end of the results section on the comparison between the decamers' and the stacks' structures.

Minor:

1. The authors use mass photometry in Fig. 1d to show the distribution of high molecular weight assemblies at different pH conditions. This could be correlated with negative stain microscopy to validate the photometry results.

We did indeed perform negative stain observations at different pHs, exactly as we did in our Jessop et al., PNAS 2021 manuscript on the LdcI polymerisation. However, the results were much less clear-cut, precisely because of the structural differences between the Adc and the LdcI stacks. Already in the LdcI manuscript, we show that while in cryo-EM (Figure 4B) the stacks are long and straight, the ns-EM grid preparation results in them being much shorter, more curved and often distorted (Figures S7 and S8). In the case of the *P. stuartii* Adc the stacks are much less stable (because of the reasons extensively described in the manuscript), and therefore in ns-EM, no matter which pH was observed, we could always see a mixture of very short and often distorted stacks, decamers and lower molecular weight species. We then tried to observe *P. stuartii* Adc at pH 4 and 5 by cryo-EM but the increasing precipitation of the stacks at lower pH precluded the obtainment of usable cryo-EM grids. These were exactly the reasons why we performed mass photometry measurements – we were not aware of this technique before and the problems that we encountered with the *P. stuartii* Adc made us realise how valuable it can be, not only as a complement to ns-EM, but also in cases as this, where ns-EM fails to provide sufficient insights.

If required, we can provide our ns-EM images for Reviewer's assessment but we would prefer not to include them in the manuscript.

2. In Fig. 1c, residues in the active site and the PLP cofactor should be labelled to guide the reader.

The figure panel has also been modified for clarity, as also suggested by Reviewer 1.

3. In Fig. 4, the meaning of "a" and "b" above the error bar is unclear and should be explained in the figure legend.

The legend of the Figure 4 has been modified.

4. "Bacteria homologous lysine and arginine decarboxylases...". I believe the authors meant homologous in the abstract.

This typo has been corrected

5. Reference 8 contains two question marks in the author name.

This typo has been corrected.

6. In the figure legend of Fig. 1, 1e is mentioned that probably existed in earlier versions of the manuscript but no longer exist.

The legend has now been modified.

7. Fig. 3c may benefit from a real close-up view to compare side chain-side chain interactions in the different assembly modes.

We would prefer to keep the current magnification for Figure 3B, as the point of the figure is to highlight the different extent of contacts between successive decamers and the difference in how extensive the interfaces are. If we zoom in, we would lose the overview of interface 1 (and 1') in LdcI. Therefore, to satisfy this Reviewer's comment, we have now included a zoom of the equivalent interfaces in Supplementary Figure 6 (new panel D), with the following legend: 'Close-up view of equivalent inter-decamer interfaces in Adc (left) and LdcI (right, termed 'interface 2' as per Figure 3B). The equivalent helices which contribute residues to the interface ($\alpha 18$ in Adc, $\alpha 16$ in LdcI) are labelled, as are stack-forming residues.'

REVIEWERS' COMMENTS:

Reviewer #2 (Remarks to the Author):

The revisions made to this MS are, by and large, excellent and responsive to the reviewer comments. The revised MS is now in quite nice shape.

One small point, the question of possible ODC activity for many ADCs, including this one, remains unresolved, and does not appear to be discussed, though I may have missed this.

The authors' rebuttal comments here are valuable in that they point to an excellent new reference in this space that discusses the parallel evolution of interconnecting polyamine pathways nicely. However, the authors have not included this discussion or this reference in their revised MS -- I would encourage them to do so.

Here is the reference. Li et al., J Biol Chem 2021 Oct;297(4):101219. doi: 10.1016/j.jbc.2021.101219.

The possibility for ODC activity of this ADC should either be tested or discussed (even if briefly) if this is beyond the domain of these studies, because this directly impacts the question of possible polyamine pathways.

Reviewer #3 (Remarks to the Author):

The authors have addressed all referees comments to my satisfaction and I recommend publication of this exciting manuscript.